biotechnology/light microscopy/cellular biology

protein tags, monovalent, covalent, DNA-PAINT imaging, superresolution

**Authors for correspondence:**
Daniel J. Nieves
e-mail: D.J.Nieves@bham.ac.uk
Katharina Gaus
e-mail: k.gaus@unsw.edu.au

†Present address: Commonwealth Scientific and Industrial Research Organisation (CSIRO), Manufacturing, Clayton, VIC 3168, Australia.

# tagPAINT: covalent labelling of genetically encoded protein tags for DNA-PAINT imaging

Daniel J. Nieves[1,2,3], Geva Hilzenrat[2,3,†], Jason Tran[2,3], Zhengmin Yang[2,3], Hugh H. MacRae[2,3], Matthew A. B. Baker[4,7], J. Justin Gooding[5,6] and Katharina Gaus[2,3]

[1]Institute of Immunology and Immunotherapy, College of Medical and Dental Sciences, University of Birmingham, Birmingham, B15 2TT, UK
[2]EMBL Australia Node in Single Molecule Science, School of Medical Sciences, [3]ARC Centre of Excellence in Advanced Molecular Imaging, [4]School of Biotechnology and Biomolecular Science, [5]School of Chemistry, and [6]Australian Centre for NanoMedicine and the ARC Centre of Excellence in Convergent Bio-Nano Science and Technology, University of New South Wales, Sydney, NSW 2052, Australia
[7]Commonwealth Scientific and Industrial Research Organisation (CSIRO), Synthetic Biology Future Science Platform, Brisbane, Australia

 DJN, 0000-0002-0873-4418; MABB, 0000-0002-5839-6904; JJG, 0000-0002-5398-0597; KG, 0000-0002-8009-9658

Recently, DNA-PAINT single-molecule localization microscopy (SMLM) has shown great promise for quantitative imaging; however, labelling strategies thus far have relied on multivalent and affinity-based approaches. Here, the covalent labelling of expressed protein tags (SNAP tag and Halo tag) with single DNA-docking strands and application of SMLM via DNA-PAINT is demonstrated. tagPAINT is then used for T-cell receptor signalling proteins at the immune synapse as a proof of principle.

## 1. Introduction

An important challenge for single-molecule localization microscopy (SMLM) for quantitative measurements is control over the stoichiometry of the label to the molecule of interest. The lack of such control can confound or alter the observed biological behaviour and states of single molecules, complexes and structures due to suboptimal labelling [1,2], cross-linking due to multivalency or label exchange [3]. To address these challenges, approaches have been developed that allow a chemically versatile stoichiometric

covalent linkage to be formed between biological molecules and fluorescent probes, in structurally defined positions [4,5]. For example, stable and stoichiometric coupling of fluorescent labels to proteins of interest has been achieved through genetically encoded affinity tags [6,7], non-canonical amino acid (ncAA) labelling [8,9] and orthogonal chemistry [10,11]. Such approaches have then been used to observe proteins at the single-molecule level [10,12].

Recently, Jungmann *et al.* demonstrated an SMLM approach using the binding/unbinding of short fluorescently conjugated DNA probes to antibodies labelled with complementary target strands, known as DNA-PAINT [13–16]. This approach was extended to determine the number of proteins/targets present in subdiffraction structures, termed qPAINT [17]. This method abrogates the uncertainty associated with the stochastic nature of fluorophore blinking and exploits *a priori* knowledge of the binding/unbinding behaviour of the probes. With this approach, good agreement was achieved between the theoretical binding/unbinding rate and the observed number of proteins. However, this approach relied on the use of probes labelled with multiple DNA target strands. Thus, multivalent interactions between proteins, multiple target strands per protein, and incomplete labelling are still challenges that need to be addressed. More recently, approaches to address this aim and minimize the linkage error include ncAA incorporation [8], affimers [18] and SOMAmers [19], which all allow 1 : 1 functionalization. However, although SOMAmers spend a long time bound, they still rely on a non-covalent interaction, and can potentially dissociate during long imaging times. Similarly, affimers are non-covalent, and sometimes require post-fixation, which may lead to off-target labelling. Also, these reagents are only available for a few protein targets to date. Finally, while ncAA incorporation does allow a covalent stoichiometric linkage, it suffers from low expression and efficiency for labelling. New approaches which allow covalent and stoichiometric labelling of a protein of interest, while maintaining a low linkage error, would thus allow robust counting of protein numbers within cell, and thus full sampling of the heterogeneity therein [16].

At present, there are a variety of methods to label proteins of interest covalently. One approach is incorporation of an enzymatically active tag, such as SNAP/CLIP tag [20] and Halo tag [21] technology. The Halo tag makes use of a chemical reaction orthogonal to eukaryotes, i.e. the dehalogenation of haloalkane ligands, thus, leading to highly specific covalent labelling of the tag, and therefore protein [21], in both live and fixed cells. Haloalkanes can be modified to bear fluorescent labels, and has been demonstrated before for SMLM in live cells using ATTO dye-modified ligands [10]. Similarly, SNAP tag, a mutant of DNA repair protein $O^6$-alkylguanine-DNA alkyltransferase, can be covalently modified using $O^6$-benzylguanine substrates (BG) and has also been demonstrated to be suitable for SMLM imaging [22]. Combining such tagging systems with DNA-PAINT imaging opens up the possibility for robust quantitative imaging of proteins within cells. This is of particular interest, as when the protein tag reaction with the ligand is successful, the achievable valency of labelling is 1 ligand : 1 tag, while also reducing the linkage error (size of tags *ca* less than 5 nm). The labelling of these proteins with DNA oligonucleotides has been demonstrated previously [23,24]; thus, the potential for using them for DNA-PAINT imaging is attractive, and has been contemporaneously explored here [25].

Here, SNAP tag and Halo tag technologies are exploited to allow 1 : 1 labelling of single proteins with a DNA-PAINT target strand (i.e. 1 protein : 1 target strand), using DNA-modified ligands to the tags. The specificity of the approach is demonstrated by targeting T-cell signalling proteins, CD3ζ and LAT, bearing the tags at the immune synapse in knockout cell lines. The potential of dual-channel tagPAINT is then explored in cells co-expressing Halo-tagged CD3ζ and SNAP-tagged LAT in T cells.

## 2. Specific imaging of SNAP- and halo-tagged proteins in knockout T-cell lines

A first important step in demonstrating the potential of this approach for DNA-PAINT imaging is to explore the specificity of the DNA ligands and their targeting to the tagged proteins. T cells when incubated on surfaces that bear T cell receptor (TCR) stimulating ligands will adhere and spread, thus, forming an area of very flat membrane at the coverglass, and thus within the total internal reflection fluorescence (TIRF) field [26]. To this end, we targeted SNAP- and Halo-tagged proteins in the immune synapse, LAT and CD3ζ (figure 1). Halo- and SNAP-tagged proteins bind covalently to a single ligand; thus, functionalization of these ligands with DNA target strands makes them amenable to DNA-PAINT imaging. Here, we used 5′-amine functionalized DNA target strands and reacted them either with excess of $O^6$-benzylguanine-NHS or Halo ligand-NHS ligands in a single step reaction (see Material and methods). Cells that were deficient in both LAT and CD3ζ were reconstituted with tagged versions of

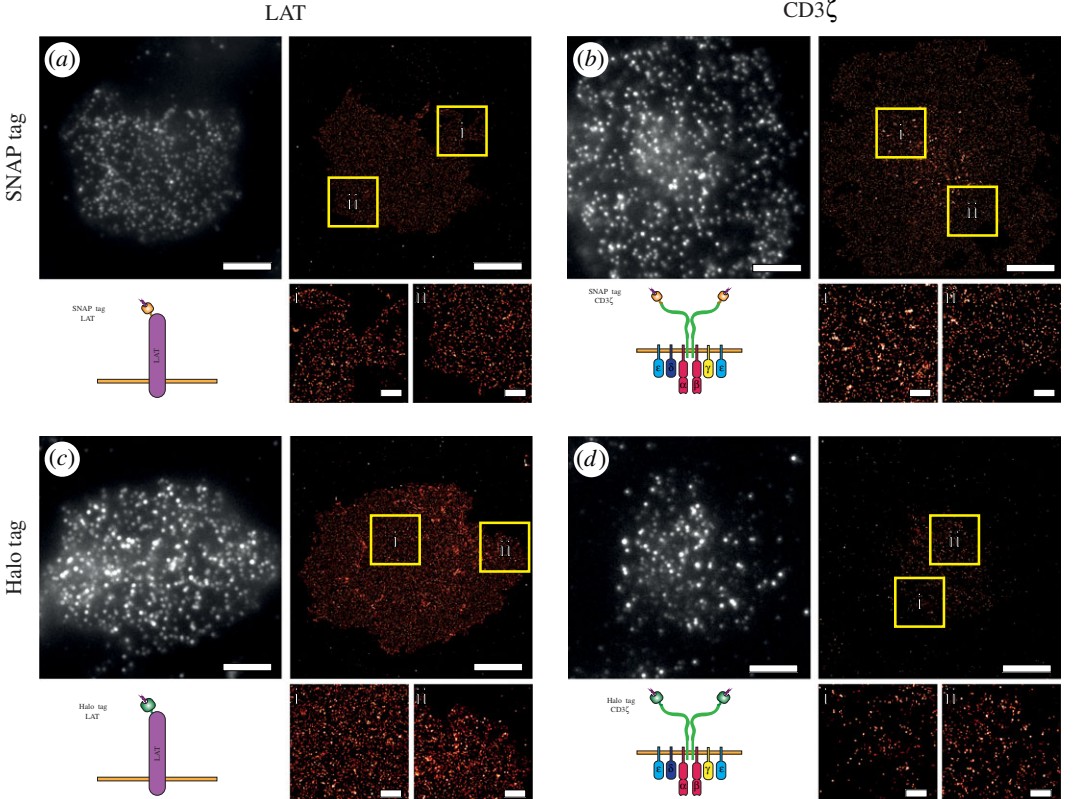

**Figure 1.** tagPAINT imaging of CD3ζ and LAT proteins. LAT and CD3ζ knockout (KO) Jurkat cells were transfected; LAT-SNAP (*a*), CD3ζ-SNAP (*b*), LAT-Halo (*c*) and CD3ζ-Halo (*d*) constructs. (*a*) LAT-SNAP transfected cells were stained with anti-SNAP-ATTO488 antibody to determine expressing cells (top left). Gaussian convolved images of positive cells imaged using DNA-PAINT (top right; 2 nM, P01-ATTO655 imager). Zoomed regions of DNA-PAINT imaging are shown (yellow boxes, i and ii). (*b*) CD3ζ-SNAP transfected cells were stained with anti-SNAP-ATTO488 antibody to determine expressing cells (top left). Gaussian convolved images of positive cells imaged using DNA-PAINT (top right; 2 nM, P01-ATTO655 imager). Zoomed regions of DNA-PAINT imaging are shown (yellow boxes, i and ii). (*c*) LAT-Halo transfected cells were stained with anti-Halo-ATTO568 antibody to determine expressing cells (top left). Gaussian convolved images of positive cells imaged using DNA-PAINT (top right; 2 nM, P03-ATTO655 imager). Zoomed regions of DNA-PAINT imaging are shown (yellow boxes, i and ii). (*d*) CD3ζ-Halo transfected cells were stained with anti-Halo-ATTO568 antibody to determine expressing cells (top left). Gaussian convolved images of positive cells imaged using DNA-PAINT (top right; 2 nM, P03-ATTO655 imager). Zoomed regions (yellow boxes, i and ii). Scale bars for all larger images are 5 µm and for zoomed images 1 µm.

these proteins via transient transfection (figure 1). To detect cells expressing the tag-bearing proteins, we used immunostaining with antibodies raised to the tag (figure 1*a*–*d*), as the tags, and the proteins they are based on, are not endogenously expressed in mammalian cells. SNAP-tagged LAT and CD3ζ-expressing cells were detected by anti-SNAP-ATTO488 and subsequently imaged using DNA-PAINT (figure 1*a*,*b*). DNA-PAINT imaging revealed the presence of punctate signal within the immune synapse, similar to that observed previously [26,27] (figure 1*a*,*b*), which is absent when cells express the SNAP tag, but have not been labelled with the DNA-SNAP-ligand (electronic supplementary material, figure S1A). The median localization precisions were 13 nm (electronic supplementary material, figure S2A) and 11 nm (electronic supplementary material, figure S2B) for LAT and CD3ζ, respectively. Halo-tagged LAT and CD3ζ-expressing cells were detected by anti-Halo-ATTO568, and again imaged by DNA-PAINT (figure 1*c*,*d*). Again, similar to SNAP tag, punctate signal is observed again at the immune synapse, and is not seen in the absence of the ligand (electronic supplementary material, figure S1B). The median localization precisions for Halo tagged (electronic supplementary material, figure S2C,D) for LAT and CD3ζ were 12 nm and 11 nm, respectively. Thus, labelling of tag for DNA-PAINT imaging appears to be highly specific.

## 3. Dual-tagPAINT imaging of CD3ζ-Halo and LAT-SNAP in the same cell

Having achieved specific targeting of the both SNAP- and Halo-tagged proteins for DNA-PAINT in isolation, we sought to demonstrate the potential for imaging the two tags within the same cell using

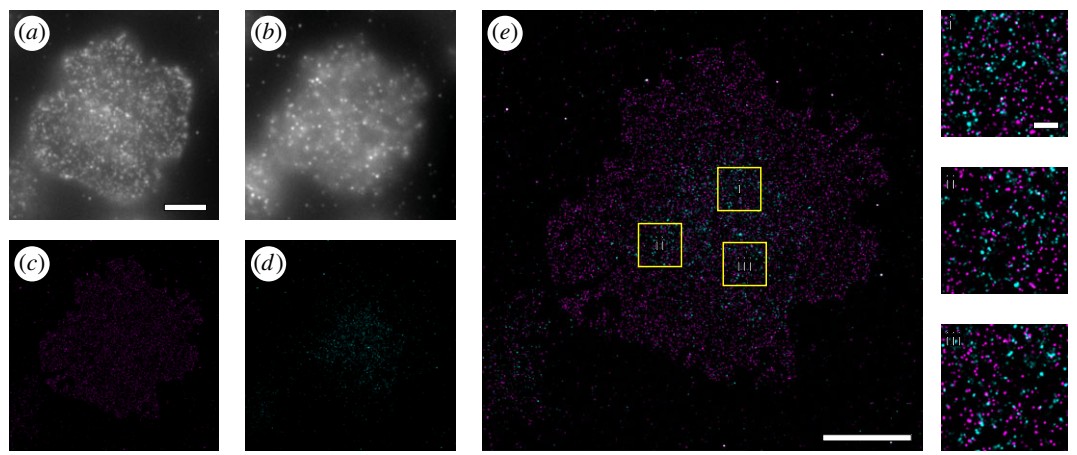

**Figure 2.** Dual-tagPAINT imaging. CD3ζ knockout (KO) Jurkat cells were transfected with both CD3ζ-Halo and LAT-SNAP constructs. Cells were stained with both (*a*) anti-SNAP-ATTO488 and (*b*) anti-Halo-ATTO568 to identify cells expressing both constructs. Convolved images of Exchange-PAINT imaging of (*c*) LAT-SNAP (2 nM, P01-ATTO655 imager) and (*d*) CD3ζ-Halo (2 nM, P01-ATTO655 imager), which are overlaid in (*e*). Zoomed regions of the overlaid images are shown from regions i–iii (yellow boxes) in *e* (right). Scale bars are 5 μm for *a*–*d* and large panel in *e*, and 500 nm for i–iii.

exchange DNA-PAINT. We used CD3ζ-knockout Jurkat T cells and expressed both a Halo-tagged CD3ζ chain and SNAP-tagged LAT protein (figure 2). These proteins are involved in the early signalling of T-cell activation and have been shown to associate and co-cluster upon engagement of the TCR [28]. Here, each protein was labelled with a ligand bearing different DNA target sequences; P01 for SNAP-LAT and P03 for CD3ζ-Halo. These sequences were chosen to avoid the possibility of cross-reaction of imaging strands across orthogonal docking strands and have been used previously for Exchange-PAINT studies [14]. Cells were stained with antibodies to the tags, as shown in figure 1, to allow identification of cells expressing both tagged proteins (figure 2*a*,*b*). Firstly, LAT-SNAP was imaged by DNA-PAINT using P01 imager (figure 2*c*), the sample was then washed with large volumes of imaging buffer, and then CD3ζ-Halo was imaged using the P03 imager (figure 2*d*). Intermixing of the two proteins is observed at the immune synapse, similar to that observed previously [28]. Thus, it is possible to orthogonally target and image different proteins within cells by employing the two tagging methods simultaneously.

## 4. Conclusion

We demonstrate the covalent conjugation of DNA target strands to SNAP and Halo tag-labelled proteins termed tagPAINT. Firstly, we demonstrated the specificity of the approach by targeting tagged CD3ζ and LAT expressed in knockout cells. Further, the potential for dual-colour imaging (both tags in the same cell) of these proteins was explored. Given that the tags only bind to a single ligand covalently, and thus will bear a single DNA target strand, this opens up the possibility for fully quantitative DNA-PAINT imaging and the opportunity to observe multiple proteins in this manner in the same cell.

## 5. Material and methods

### 5.1. Plasmids

For the expression of Halo tag-labelled human CD3ζ, the EGFP gene in the vector pEGFP-N1 was exchanged with the Halo tag gene after restriction digest, generating an empty back bone where human CD3ζ could be inserted. For SNAP-CD3ζ, the Halo tag was subsequently replaced with SNAP gene, using the AgeI and NotI sites. For SNAP-LAT, human CD3ζ of the SNAP-CD3ζ construct was replaced with human LAT.

### 5.2. Synthesis of Halo tag-DNA ligands

5′amino modified DNA-docking strands (either P01: 5′- TTATACATCTA-3′ or P03: 5′- TTTCTTCATTA-3′) were diluted in 10 mM sodium phosphate buffer pH 6.8, supplemented with 5 mM EDTA, to a final

concentration of 1 mM. N-hydroxysuccinimidyl ester functionalized Halo tag ligand (NHS-HL) was freshly reconstituted in dry DMSO to a final concentration of 50 mM (unused NHS-HL was aliquoted and stored at −80°C). NHS-HL was diluted 10 times by adding 5′amino modified DNA-docking strands and mixing thoroughly. The reaction was left for 1 h at room temperature. The reaction product, HL functionalized with a DNA-docking strand, was purified from excess unreacted NHS-HL by size exclusion chromatography, with 10 mM Tris supplemented with 1 mM EDTA as the mobile phase. Purified HL-DNA-docking strands were aliquoted and stored at −20°C until use (final concentration approx. 100–200 µM).

## 5.3. Synthesis of SNAP tag-DNA ligands

5′amino modified DNA-docking strands were diluted in 10 mM sodium phosphate buffer pH 6.8, supplemented with 5 mM EDTA, to a final concentration of 1 mM. N-hydroxysuccinimidyl ester functionalized SNAP ligand ($O^6$-benzylguanine) was freshly reconstituted in dry DMSO to a final concentration of 50 mM (unused NHS-$O^6$-benzylguanine was aliquoted and stored at −80°C). NHS-$O^6$-benzylguanine was diluted 10 times by adding 5′amino modified DNA-docking strands and mixing thoroughly. The reaction was left for 1 h at room temperature. The reaction product, $O^6$-benzylguanine functionalized with a DNA-docking strand, was purified from excess unreacted NHS-$O^6$-benzylguanine by size exclusion chromatography, with 10 mM Tris supplemented with 1 mM EDTA as the mobile phase. Purified HL-DNA-docking strands were aliquoted and stored at −20°C until use (final concentration approx. 100–200 µM).

## 5.4. Transfection and fixation of Jurkat T cells

E6.1 Jurkat T cells were transfected with Halo or SNAP constructs using an Invitrogen Neon Electroporation Transfection System (Life Technologies Pty Ltd), using three pulses of 1350 V lasting 10 ms. The cells were left to recover after transfection in RPMI medium without Phenol Red (6040, GIBCO) supplemented 20% (v/v) fetal bovine serum. Before seeding, cells were pelleted by centrifugation, washed once with PBS and then resuspended in PBS. Cells were then used for seeding onto coverslips coated with 10 µg ml$^{-1}$ anti-human CD3ε (activating) for 10 min at 37°C with 5% CO$_2$, after which non-adherent cells were washed away with PBS and then fixed with freshly prepared warm 4% (w/v) PFA in PBS for 10 min. Fixative was then washed away with PBS and then cells were permeabilized for labelling with 0.1% (v/v) Triton X-100 in PBS for 3 min.

## 5.5. Immunostaining for SNAP and Halo tags

Cells expressing Halo and SNAP tags were processed for staining in an identical manner. After fixation, cells were permeabilized with PBS supplemented with 0.2% Triton-X100 for 3 min, and then subsequently incubated with 50 mg ml$^{-1}$ BSA in PBS for 30 min. After which, the cells were stained with either anti-SNAP-ATTO488 or anti-Halo-ATTO568 (or both; figure 2) at a final concentration of 1 µg ml$^{-1}$ in 50 mg ml$^{-1}$ BSA in PBS for 30 min. Cells were then washed with PBS and post-fixed using 4% PFA in PBS.

## 5.6. Labelling with SNAP and Halo DNA ligands

Both DNA-functionalized SNAP and Halo ligands were incubated with fixed and immunostained cell samples at a final concentration of 5 µM in PBS supplemented with 0.2% Tween-20 (PBST) for 10 min. The samples were then vigorously washed with 1 ml of PBST several times, to remove any non-specifically adsorbed ligand. Finally, the samples were incubated with gold nanorods (Nanopartz) in PBST for 10 min before mounting for tagPAINT imaging.

## 5.7. tagPAINT imaging

Prior to imaging labelled cells, glass coverslips were mounted into a Chamlide chamber and freshly prepared imaging strands (2 nM ATTO655 P01 or 2 nM ATTO655 P03) in PBST supplemented with 500 mM NaCl were added to the chamber. For tagPAINT imaging, ATTO655 imager binding was acquired with the 642 nm (0.075 kW cm$^{-2}$) laser lines, respectively. For standard tagPAINT imaging, an integration time of 80 ms was used, with a TIRF angle of 66.90° (penetration depth = 110 nm), with

40 000 frames acquired. Images were acquired as 512 × 512 pixel images with a pixel size of 97 nm. For Exchange-PAINT imaging (figure 2), the same imaging parameters as above were used, but in between each image series, the imager solution was exchanged by washing with at least 10 × 1 ml PBST supplemented with 500 mM NaCl, and then adding in the second imager solution.

## 5.8. tagPAINT processing and image convolution

tagPAINT images were processed using Zeiss Zen Black software. The position of bound imaging strands in the acquisition was determined by Gaussian fitting, using a peak mask radius size of 6 pixels and a signal to noise ratio cut-off of 8. The localization data were then drift corrected using the point patterns generated from the localization of gold nanorod fiducials within the field of view using the Zeiss Zen Black software drift correction. The point pattern data were then convolved with a Gaussian kernel using the ThunderSTORM plugin in Fiji, with a pixel size of 9.7 nm. The median precision values were extracted from the distribution of precision values (electronic supplementary material, figure S2) given in the output of the Zen processing.

Data accessibility. The data used to create this images and figures in the manuscript can be found on Figshare, collection 'tagPAINT data' [31]: https://doi.org/10.6084/m9.figshare.c.2928956.

Authors' contributions. D.J.N. conceived the study, designed the experiments, synthesized DNA tagging reagents, acquired data, wrote and drafted the manuscript, G.H. aided in experimental design, method optimization and analysed data, J.T. aided in experimental design, imaging and labelling/method optimization, Z.Y. cloned and generated constructs, H.H.M. cloned and generated constructs, M.A.B.B. aided in experimental design and drafting of the manuscript, J.J.G. aided in ligand synthesis optimization and drafting of the manuscript, K.G. aided in experimental design and aided in writing and drafting of the manuscript. All authors gave final approval for publication.

Competing interests. We declare we have no competing interests.

Funding. We acknowledge technical assistance by the BioMedical Imaging Facility, University of New South Wales. K.G. acknowledges funding from the ARC Centre of Excellence in Advanced Molecular Imaging (CE140100011 to K.G.) and National Health and Medical Research Council of Australia (1059278 and 1037320 to K.G.). K.G also acknowledges the Australian Research Council (LP140100967 and DP130100269). J.J.G. acknowledges funding from the ARC Centre of Excellence in Convergent Bio-Nano Science and Technology (CE140100036), the Australian Research Council for an Australian Laureate Fellowship (FL150100060) and a National Health and Medical Research Council of Australia programme grant (1091261). G.H. acknowledges the supported by an Australian Government Research Training Program (RTP) Scholarship.

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
