## [Reviewer comments · Royal Society Open Science]

Review History

RSOS-191268.R0 (Original submission)

Review form: Reviewer 1

Is the manuscript scientifically sound in its present form?

Yes

Are the interpretations and conclusions justified by the results?

Yes

Is the language acceptable?

Yes

Do you have any ethical concerns with this paper?

No

Have you any concerns about statistical analyses in this paper?

No

Recommendation?

Accept with minor revision (please list in comments)

Comments to the Author(s)

This is work performed by an expert group in single molecule localisation microscopy (SMLM) of the immune synapse, specifically, T-Cell receptor signalling. The author use proteins of the T-Cell receptor complex to create fusion constructs with two commonly used different enzymatic labelling systems called Halo-tag and SNAP-tag. These constructs are then transfected into Jurkat cells and imaged using DNA-oligomer coupled SNAP-tag or Halo-tag ligands. DNA-PAINT imaging was then performed using fluorophore-coupled complementary oligomers. The authors show single channel and dual channel data and some controls and statistics. Overall a solid piece of work that can be published after a few minor changes.

Minor points:

1) If the authors want to state that: "...a chemically versatile stoichiometric covalent linkage...", they have the burden to demonstrate stoichiometric labelling. Others have shown that this is neither for the Halo nor the SNAP tag readily achieved
<https://www.biorxiv.org/content/10.1101/582668v1>. I suggest a change of wording by elimination of the word "stoichiometric".

2) The authors should cite Schlichthaerle et al., 2019 Angewandte Chemie doi: 10.1002/anie.201905685 in which the technique the authors present here was published before.

Review form: Reviewer 2**Is the manuscript scientifically sound in its present form?**

Yes

Are the interpretations and conclusions justified by the results?

Yes

Is the language acceptable?

Yes

Do you have any ethical concerns with this paper?

No

Have you any concerns about statistical analyses in this paper?

No

Recommendation?

Accept with minor revision (please list in comments)

Comments to the Author(s)

In the manuscript "tagPAINT: covalent labelling of genetically encoded protein tags for DNA-PAINT imaging" by Nieves et al, the authors describe a methodology to super-resolve SNAP or Halo-tagged proteins using DNA-PAINT. They achieve this by functionalizing protein(s) of

interest with SNAP or Halo tag, after which they introduce ssDNAs conjugated to the respective ligands, and fluorescently labelled complementary ssDNA. With this method, they show single- and dual-colour measurements of LAT or CD3-zeta proteins with sub 20 nm localization precision.

The manuscript is concisely written, and the text is mostly clearly structured. The authors clearly show their methodology is working and provide strong results.

The manuscript could be further strengthened by adding a control experiment in which a ssDNA is ligated to the SNAP or Halo tag, after which a non-complementary but fluorescently labelled ssDNA is introduced. This experiment would provide insight on “cross-binding” of DNA-PAINT. This information would be important to accurately assess the results presented in Fig. 2.

I am a bit confused about the methodology underlying figure S2. Here, the authors provide localization precision information about the single molecule localizations. However, instead of providing actual localization accuracies (which can easily be extracted by using software such as ThunderSTORM or SMAP), they provide a ‘best-case’ localization precision based on photon numbers, background, and pixel size. This is all based on theoretical calculation, and might not hold true for real-life scenarios.

Other remarks:

- Throughout the introduction, the authors stress the importance of the quantitative possibilities of DNA-PAINT (qPAINT), but they never show this with their own data
- Methods, line 43-44, the authors discuss a certain TIRF angle. This value would make more sense expressed as how far the HiLo/TIRF field (approximately) excites into the sample.
- The authors should comment on why Halo-tag-DNA-PAINT provides better localization precision compared to SNAP-tag-DNA-PAINT (the histograms in Fig S2 are much broader, even if the median value is mostly unchanged). The SNAP tag is slightly smaller than the Halo tag (physically), which would suggest to provide better localization precision compared to the Halo-tag-DNA-PAINT.
- Results, line 31,50-51: “observed at the immune synapse”: this is not clear from the image for the untrained eye. Maybe the authors can add a sentence for further explanation or highlight the finding in the images.
- Figure 1, caption: I think ‘top left’ and ‘top right’ is switched around throughout the caption. The antibody-labelled image is shown in top-left, while the DNA-PAINT labeled image is shown top right.

Decision letter (RSOS-191268.R0)

14-Oct-2019

Dear Dr Nieves

On behalf of the Editors, I am pleased to inform you that your Manuscript RSOS-191268 entitled "tagPAINT: covalent labelling of genetically encoded protein tags for DNA-PAINT imaging" has been accepted for publication in Royal Society Open Science subject to minor revision in accordance with the referee suggestions. Please find the referees' comments at the end of this email.

The reviewers and handling editors have recommended publication, but also suggest some minor revisions to your manuscript. Therefore, I invite you to respond to the comments and revise your manuscript.

- Ethics statement

- Data accessibility

<http://datadryad.org/submit?journalID=RSOS&manu=RSOS-191268>

- Competing interests

- Authors' contributions

- Acknowledgements

- Funding statement

Because the schedule for publication is very tight, it is a condition of publication that you submit the revised version of your manuscript before 23-Oct-2019. Please note that the revision deadline will expire at 00.00am on this date. If you do not think you will be able to meet this date please let me know immediately.

Kind regards,
Anita Kristiansen
Editorial Coordinator
Royal Society Open Science
openscience@royalsociety.org

on behalf of Dr Andrew Turberfield (Associate Editor) and Pietro Cicuta (Subject Editor)
openscience@royalsociety.org

Reviewer comments to Author:
Reviewer: 1

Comments to the Author(s)

This is work performed by an expert group in single molecule localisation microscopy (SMLM) of the immune synapse, specifically, T-Cell receptor signalling. The author use proteins of the T-Cell receptor complex to create fusion constructs with two commonly used different enzymatic labelling systems called Halo-tag and SNAP-tag. These constructs are then transfected into Jurkat cells and imaged using DNA-oligomer coupled SNAP-tag or Halo-tag ligands. DNA-PAINT imaging was then performed using fluorophore-coupled complementary oligomers. The authors show single channel and dual channel data and some controls and statistics. Overall a solid piece of work that can be published after a few minor changes.

Minor points:

1) If the authors want to state that: "...a chemically versatile stoichiometric covalent linkage...", they have the burden to demonstrate stoichiometric labelling. Others have shown that this is neither for the Halo nor the SNAP tag readily achieved
<https://www.biorxiv.org/content/10.1101/582668v1>. I suggest a change of wording by elimination of the word "stoichiometric".

2) The authors should cite Schlichthaerle et al., 2019 *Angewandte Chemie* doi: 10.1002/anie.201905685 in which the technique the authors present here was published before.

Reviewer: 2

Comments to the Author(s)

In the manuscript “tagPAINT: covalent labelling of genetically encoded protein tags for DNA-PAINT imaging” by Nieves et al, the authors describe a methodology to super-resolve SNAP or Halo-tagged proteins using DNA-PAINT. They achieve this by functionalizing protein(s) of interest with SNAP or Halo tag, after which they introduce ssDNAs conjugated to the respective ligands, and fluorescently labelled complementary ssDNA. With this method, they show single- and dual-colour measurements of LAT or CD3-zeta proteins with sub 20 nm localization precision.

The manuscript is concisely written, and the text is mostly clearly structured. The authors clearly show their methodology is working and provide strong results.

The manuscript could be further strengthened by adding a control experiment in which a ssDNA is ligated to the SNAP or Halo tag, after which a non-complementary but fluorescently labelled ssDNA is introduced. This experiment would provide insight on “cross-binding” of DNA-PAINT. This information would be important to accurately assess the results presented in Fig. 2.

I am a bit confused about the methodology underlying figure S2. Here, the authors provide localization precision information about the single molecule localizations. However, instead of providing actual localization accuracies (which can easily be extracted by using software such as ThunderSTORM or SMAP), they provide a ‘best-case’ localization precision based on photon numbers, background, and pixel size. This is all based on theoretical calculation, and might not hold true for real-life scenarios.

Other remarks:

- Throughout the introduction, the authors stress the importance of the quantitative possibilities of DNA-PAINT (qPAINT), but they never show this with their own data
- Methods, line 43-44, the authors discuss a certain TIRF angle. This value would make more sense expressed as how far the HiLo/TIRF field (approximately) excites into the sample.
- The authors should comment on why Halo-tag-DNA-PAINT provides better localization precision compared to SNAP-tag-DNA-PAINT (the histograms in Fig S2 are much broader, even if the median value is mostly unchanged). The SNAP tag is slightly smaller than the Halo tag (physically), which would suggest to provide better localization precision compared to the Halo-tag-DNA-PAINT.
- Results, line 31,50-51: “observed at the immune synapse”: this is not clear from the image for the untrained eye. Maybe the authors can add a sentence for further explanation or highlight the finding in the images.
- Figure 1, caption: I think ‘top left’ and ‘top right’ is switched around throughout the caption. The antibody-labelled image is shown in top-left, while the DNA-PAINT labeled image is shown top right.

Author's Response to Decision Letter for (RSOS-191268.R0)

See Appendix A.

Decision letter (RSOS-191268.R1)

04-Nov-2019

Dear Dr Nieves:

On behalf of the Editors, I am pleased to inform you that your Manuscript RSOS-191268.R1 entitled "tagPAINT: covalent labelling of genetically encoded protein tags for DNA-PAINT imaging" has been accepted for publication in Royal Society Open Science subject to minor revision in accordance with the referee suggestions. Please find the referees' comments at the end of this email.

The reviewers and Subject Editor have recommended publication, but also suggest some minor revisions to your manuscript. Therefore, I invite you to respond to the comments and revise your manuscript.

- Ethics statement

- Data accessibility

If you wish to submit your supporting data or code to Dryad (<http://datadryad.org/>), or modify your current submission to dryad, please use the following link:
<http://datadryad.org/submit?journalID=RSOS&manu=RSOS-191268.R1>

- Competing interests

- Authors' contributions

- Acknowledgements

- Funding statement

Because the schedule for publication is very tight, it is a condition of publication that you submit the revised version of your manuscript before 13-Nov-2019. Please note that the revision deadline will expire at 00.00am on this date. If you do not think you will be able to meet this date please let me know immediately.

Supplementary files will be published alongside the paper on the journal website and posted on

the online figshare repository (<https://figshare.com>). The heading and legend provided for each supplementary file during the submission process will be used to create the figshare page, so please ensure these are accurate and informative so that your files can be found in searches. Files on figshare will be made available approximately one week before the accompanying article so that the supplementary material can be attributed a unique DOI.

on behalf of Dr Andrew Turberfield (Associate Editor) and Pietro Cicuta (Subject Editor)
openscience@royalsociety.org

Associate Editor Comments to Author (Dr Andrew Turberfield):

:

The authors have addressed the reviewers' comments successfully. I recommend one further change in response to Reviewer 1's objection to the ambiguous use of the word 'stoichiometry'. In the following paragraph:

"Here, SNAP tag and Halo Tag technologies are exploited to allow stoichiometric labelling of single proteins with a DNA PAINT target strand (i.e., 1 protein: 1 target strand), ...". Please replace "stoichiometric" with "1:1", which does not have such a loaded meaning.

Author's Response to Decision Letter for (RSOS-191268.R1)

See Appendix B.

Decision letter (RSOS-191268.R2)

12-Nov-2019

Dear Dr Nieves,

It is a pleasure to accept your manuscript entitled "tagPAINT: covalent labelling of genetically encoded protein tags for DNA-PAINT imaging" in its current form for publication in Royal Society Open Science. The comments of the reviewer(s) who reviewed your manuscript are included at the foot of this letter.

You can expect to receive a proof of your article in the near future. Please contact the editorial office (openscience_proofs@royalsociety.org) and the production office

(openscience@royalsociety.org) to let us know if you are likely to be away from e-mail contact -- if you are going to be away, please nominate a co-author (if available) to manage the proofing process, and ensure they are copied into your email to the journal.

Kind regards,
Andrew Dunn
Senior Publishing Editor
Royal Society Open Science Editorial Office
Royal Society Open Science
openscience@royalsociety.org

on behalf of Dr Andrew Turberfield (Associate Editor) and Pietro Cicuta (Subject Editor)
openscience@royalsociety.org

Appendix A

To the Subject Editor,

We would first like to thank you for allowing us to revise our manuscript (RSOS-181268), in the light of the reviewers' comments, for publication in Royal Society Open Science. We would also thank the reviewers for their excellent and helpful comments, which we have used to improve the readability and quality of the manuscript. Below is our response to those points raised by the reviewers (in red)

Reviewer: 1

Comments to the Author(s)

This is work performed by an expert group in single molecule localisation microscopy (SMLM) of the immune synapse, specifically, T-Cell receptor signalling. The author use proteins of the T-Cell receptor complex to create fusion constructs with two commonly used different enzymatic labelling systems called Halo-tag and SNAP-tag. These constructs are then transfected into Jurkat cells and imaged using DNA-oligomer coupled SNAP-tag or Halo-tag ligands. DNA-PAINT imaging was then performed using fluorophore-coupled complementary oligomers. The authors show single channel and dual channel data and some controls and statistics. Overall a solid piece of work that can be published after a few minor changes.

Minor points:

1) If the authors want to state that: "...a chemically versatile stoichiometric covalent linkage...", they have the burden to demonstrate stoichiometric labelling. Others have shown that this is neither for the Halo nor the SNAP tag readily achieved <https://www.biorxiv.org/content/10.1101/582668v1>. I suggest a change of wording by elimination of the word "stoichiometric".

With stoichiometric labelling, we meant that under ideal conditions, a pre-set ratio of labels to protein-of-interest can be achieved. For antibody staining, for example, even under optimised conditions, one cannot guarantee that only one antibody binds to the target protein. With the Halo and SNAP tags, when the reaction of the tags with these ligands is successful, only one ligand per protein is achievable, thus a 1:1 stoichiometry (as cited in text already). We have amended the text to make clearer this distinction between "labelling efficiency" and "stoichiometry/valency" (lines: 86-87).

As the reviewer rightly points out it is not easy to achieve a 100% labelling efficiency with these tags (or rather measure 100% labelling efficiency), as demonstrated in the paper cited above. We also do not claim 100% labelling efficiency here, but that the tags are a means to control valency, critical for future quantitative studies. It should be noted in Vermal *et al.*, efficiency measurements were taken with SMLM (dSTORM) of dye-labelled ligands, and reported a range of 60-80%. Unfortunately, it is a longstanding issue that dSTORM suffers inherently from not being able to observe all the dyes present within the sample, due to photobleaching in the high excitation power reduction step. Therefore, the labelling efficiency could be higher than reported in these dSTORM measurements.

2) The authors should cite Schlichthaerle et al., 2019 *Angewandte Chemie* doi: 10.1002/anie.201905685 in which the technique the authors present here was published before.

This contemporaneous work is now cited in the manuscript (lines: 89-90).

Reviewer: 2

Comments to the Author(s)

In the manuscript "tagPAINT: covalent labelling of genetically encoded protein tags for DNA-PAINT imaging" by Nieves et al, the authors describe a methodology to super-resolve SNAP or Halo-tagged proteins using DNA-PAINT. They achieve this by functionalizing protein(s) of interest with SNAP or Halo tag, after which they introduce ssDNAs conjugated to the respective ligands, and fluorescently labelled complementary ssDNA. With this method, they show single- and dual-colour measurements of LAT or CD3-zeta proteins with sub 20 nm localization precision.

The manuscript is concisely written, and the text is mostly clearly structured. The authors clearly show their methodology is working and provide strong results.

The manuscript could be further strengthened by adding a control experiment in which a ssDNA is ligated to the SNAP or Halo tag, after which a non-complementary but fluorescently labelled ssDNA is introduced. This experiment would provide insight on "cross-binding" of DNA-PAINT. This information would be important to accurately assess the results presented in Fig. 2.

Firstly, our sequences used were omitted from the original manuscript text in error and have now been added to the methods section (lines: 161-162).

The reviewer makes an excellent point about cross reactivity. This has been addressed before by the Jungmann *et al.*, in their Exchange-PAINT paper (Nature Methods, 11, 313–318, 2014), and our sequences (P01 and P03) were two of those chosen for imaging orthologous sequence docking sites on a DNA origami in that work. This was also our motivation for selecting these sequences, i.e., to remove the possibility of cross-binding. Therefore, the text has been amended to make our motivations clear (lines: 137-139).

I am a bit confused about the methodology underlying figure S2. Here, the authors provide localization precision information about the single molecule localizations. However, instead of providing actual localization accuracies (which can easily be extracted by using software such as ThunderSTORM or SMAP), they provide a 'best-case' localization precision based on photon numbers, background, and pixel size. This is all based on theoretical calculation and might not hold true for real-life scenarios.

These precision values, and the accompanying histograms were generated from the ThunderSTORM output tables. The main text and methods have been amended to explain how the precision values were derived, and the median precision values (i.e., the median value of all localised molecules within the cell) are now reported in text (lines: 113-118, 217-218).

Other remarks:

- Throughout the introduction, the authors stress the importance of the quantitative possibilities of DNA-PAINT (qPAINT), but they never show this with their own data

qPAINT was attempted with this system, but not presented here, as the calibration of the data, especially in a biological sample was very challenging. Recently, qPAINT has been significantly improved to remove the need to for in sample calibrations (here: <https://doi.org/10.1021/acs.nanolett.9b03546>), like DNA origamis. We mention qPAINT in the introduction, as it measures the number of docking strands in the sample, not proteins. Thus, being able to control valency, and also permanently label proteins is potentially critical for biological measurements with qPAINT, and was the motivation for the work.

- Methods, line 43-44, the authors discuss a certain TIRF angle. This value would make more sense expressed as how far the HiLo/TIRF field (approximately) excites into the sample.

Penetration depth of the TIRF field added to the methods section (lines: 206-207).

- The authors should comment on why Halo-tag-DNA-PAINT provides better localization precision compared to SNAP-tag-DNA-PAINT (the histograms in Fig S2 are much broader, even if the median value is mostly unchanged). The SNAP tag is slightly smaller than the Halo tag (physically), which would suggest to provide better localization precision compared to the Halo-tag-DNA-PAINT.

We have improved the segmentation of the cell data in the images, and have now amended the precision histograms (Figure S2). In the original Figure S2 it seemed indeed that Halo was narrower than the SNAP. Now the histograms are very similar for both Halo and SNAP, with a broadened CD3zeta-Halo distribution around the median value. This broadening is like due to cell morphology at the coverslip as it is not as spread as the other examples. The shoulders on these precision data, from approximately 15-30 nm, for all histograms are likely due to proteins that are higher in the TIRF field as these proteins are shuttled to and from the synapse during activation, and may be from proteins about to be or being endocytosed.

- Results, line 31,50-51: "observed at the immune synapse": this is not clear from the image for the untrained eye. Maybe the authors can add a sentence for further explanation or highlight the finding in the images.

Text has been amended to give more detail to non-expert audience on how the synapse is generated at the coverslip (with reference to where this method has been employed for SMLM before) and how it lies within the TIRF field (lines: 98-100).

- Figure 1, caption: I think 'top left' and 'top right' is switched around throughout the caption. The antibody-labelled image is shown in top-left, while the DNA-PAINT labelled image is shown top right.

Now corrected (Figure 1 Caption).

Appendix B

To the Subject Editor,

We would first like to thank you for allowing us to revise our manuscript (RSOS-181268), in the light of the reviewers' comments, for publication in Royal Society Open Science. We would also thank the reviewers for their excellent and helpful comments, which we have used to improve the readability and quality of the manuscript. Below is our response to those points raised by the reviewers (in red)

Reviewer: 1

Comments to the Author(s)

This is work performed by an expert group in single molecule localisation microscopy (SMLM) of the immune synapse, specifically, T-Cell receptor signalling. The author use proteins of the T-Cell receptor complex to create fusion constructs with two commonly used different enzymatic labelling systems called Halo-tag and SNAP-tag. These constructs are then transfected into Jurkat cells and imaged using DNA-oligomer coupled SNAP-tag or Halo-tag ligands. DNA-PAINT imaging was then performed using fluorophore-coupled complementary oligomers. The authors show single channel and dual channel data and some controls and statistics. Overall a solid piece of work that can be published after a few minor changes.

Minor points:

1) If the authors want to state that: "...a chemically versatile stoichiometric covalent linkage...", they have the burden to demonstrate stoichiometric labelling. Others have shown that this is neither for the Halo nor the SNAP tag readily achieved <https://www.biorxiv.org/content/10.1101/582668v1>. I suggest a change of wording by elimination of the word "stoichiometric".

With stoichiometric labelling, we meant that under ideal conditions, a pre-set ratio of labels to protein-of-interest can be achieved. For antibody staining, for example, even under optimised conditions, one cannot guarantee that only one antibody binds to the target protein. With the Halo and SNAP tags, when the reaction of the tags with these ligands is successful, only one ligand per protein is achievable, thus a 1:1 stoichiometry (as cited in text already). We have amended the text to make clearer this distinction between "labelling efficiency" and "stoichiometry/valency" (lines: 86-87).

As the reviewer rightly points out it is not easy to achieve a 100% labelling efficiency with these tags (or rather measure 100% labelling efficiency), as demonstrated in the paper cited above. We also do not claim 100% labelling efficiency here, but that the tags are a means to control valency, critical for future quantitative studies. It should be noted in Vermal *et al.*, efficiency measurements were taken with SMLM (dSTORM) of dye-labelled ligands, and reported a range of 60-80%. Unfortunately, it is a longstanding issue that dSTORM suffers inherently from not being able to observe all the dyes present within the sample, due to photobleaching in the high excitation power reduction step. Therefore, the labelling efficiency could be higher than reported in these dSTORM measurements.

2) The authors should cite Schlichthaerle et al., 2019 *Angewandte Chemie* doi: 10.1002/anie.201905685 in which the technique the authors present here was published before.

This contemporaneous work is now cited in the manuscript (lines: 89-90).

Reviewer: 2

Comments to the Author(s)

In the manuscript “tagPAINT: covalent labelling of genetically encoded protein tags for DNA-PAINT imaging” by Nieves et al, the authors describe a methodology to super-resolve SNAP or Halo-tagged proteins using DNA-PAINT. They achieve this by functionalizing protein(s) of interest with SNAP or Halo tag, after which they introduce ssDNAs conjugated to the respective ligands, and fluorescently labelled complementary ssDNA. With this method, they show single- and dual-colour measurements of LAT or CD3-zeta proteins with sub 20 nm localization precision.

The manuscript is concisely written, and the text is mostly clearly structured. The authors clearly show their methodology is working and provide strong results.

The manuscript could be further strengthened by adding a control experiment in which a ssDNA is ligated to the SNAP or Halo tag, after which a non-complementary but fluorescently labelled ssDNA is introduced. This experiment would provide insight on “cross-binding” of DNA-PAINT. This information would be important to accurately assess the results presented in Fig. 2.

Firstly, our sequences used were omitted from the original manuscript text in error and have now been added to the methods section (lines: 161-162).

The reviewer makes an excellent point about cross reactivity. This has been addressed before by the Jungmann *et al.*, in their Exchange-PAINT paper (Nature Methods, 11, 313–318, 2014), and our sequences (P01 and P03) were two of those chosen for imaging orthologous sequence docking sites on a DNA origami in that work. This was also our motivation for selecting these sequences, i.e., to remove the possibility of cross-binding. Therefore, the text has been amended to make our motivations clear (lines: 137-139).

I am a bit confused about the methodology underlying figure S2. Here, the authors provide localization precision information about the single molecule localizations. However, instead of providing actual localization accuracies (which can easily be extracted by using software such as ThunderSTORM or SMAP), they provide a 'best-case' localization precision based on photon numbers, background, and pixel size. This is all based on theoretical calculation and might not hold true for real-life scenarios.

These precision values, and the accompanying histograms were generated from the ThunderSTORM output tables. The main text and methods have been amended to explain how the precision values were derived, and the median precision values (i.e., the median value of all localised molecules within the cell) are now reported in text (lines: 113-118, 217-218).

Other remarks:

- Throughout the introduction, the authors stress the importance of the quantitative possibilities of DNA-PAINT (qPAINT), but they never show this with their own data

qPAINT was attempted with this system, but not presented here, as the calibration of the data, especially in a biological sample was very challenging. Recently, qPAINT has been significantly improved to remove the need to for in sample calibrations (here: <https://doi.org/10.1021/acs.nanolett.9b03546>), like DNA origamis. We mention qPAINT in the introduction, as it measures the number of docking strands in the sample, not proteins. Thus, being able to control valency, and also permanently label proteins is potentially critical for biological measurements with qPAINT, and was the motivation for the work.

- Methods, line 43-44, the authors discuss a certain TIRF angle. This value would make more sense expressed as how far the HiLo/TIRF field (approximately) excites into the sample.

Penetration depth of the TIRF field added to the methods section (lines: 206-207).

- The authors should comment on why Halo-tag-DNA-PAINT provides better localization precision compared to SNAP-tag-DNA-PAINT (the histograms in Fig S2 are much broader, even if the median value is mostly unchanged). The SNAP tag is slightly smaller than the Halo tag (physically), which would suggest to provide better localization precision compared to the Halo-tag-DNA-PAINT.

We have improved the segmentation of the cell data in the images, and have now amended the precision histograms (Figure S2). In the original Figure S2 it seemed indeed that Halo was narrower than the SNAP. Now the histograms are very similar for both Halo and SNAP, with a broadened CD3zeta-Halo distribution around the median value. This broadening is likely due to cell morphology at the coverslip as it is not as spread as the other examples. The shoulders on these precision data, from approximately 15-30 nm, for all histograms are likely due to proteins that are higher in the TIRF field as these proteins are shuttled to and from the synapse during activation, and may be from proteins about to be or being endocytosed.

- Results, line 31,50-51: “observed at the immune synapse”: this is not clear from the image for the untrained eye. Maybe the authors can add a sentence for further explanation or highlight the finding in the images.

Text has been amended to give more detail to non-expert audience on how the synapse is generated at the coverslip (with reference to where this method has been employed for SMLM before) and how it lies within the TIRF field (lines: 98-100).

- Figure 1, caption: I think ‘top left’ and ‘top right’ is switched around throughout the caption. The antibody-labelled image is shown in top-left, while the DNA-PAINT labelled image is shown top right.

Now corrected (Figure 1 Caption).